# Prevalence of Diabetes and Its Association with Atherosclerotic Cardiovascular Disease Risk in Patients with Familial Hypercholesterolemia: An Analysis from the Hellenic Familial Hypercholesterolemia Registry (HELLAS-FH)

**DOI:** 10.3390/ph16010044

**Published:** 2022-12-28

**Authors:** Chrysoula Boutari, Christos V. Rizos, Michalis Doumas, George Liamis, Ioannis Skoumas, Loukianos Rallidis, Anastasia Garoufi, Genovefa Kolovou, Konstantinos Tziomalos, Emmanouil Skalidis, Vasileios Kotsis, George Sfikas, Vaia Lambadiari, Panagiotis Anagnostis, Eleni Bilianou, Georgia Anastasiou, Iosif Koutagiar, Estela Kiouri, Achilleas Attilakos, Vana Kolovou, Evangelos Zacharis, Christina Antza, Evangelos Liberopoulos

**Affiliations:** 12nd Propedeutic Department of Internal Medicine, Medical School, Aristotle University of Thessaloniki, Hippokration General Hospital, 54642 Thessaloniki, Greece; 2Department of Internal Medicine, Medical School, University of Ioannina, 45110 Ioannina, Greece; 3Cardiology Clinic, Hippokration General Hospital, 54643 Athens, Greece; 4Department of Cardiology, Medical School, Attikon University General Hospital, National and Kapodistrian University of Athens, 12462 Athens, Greece; 52nd Department of Pediatrics, Medical School, National and Kapodistrian University of Athens, “P. & A. Kyriakou” Children’s Hospital, 15452 Athens, Greece; 6Cardiometabolic Center, Lipid Clinic, LA Apheresis Unit, Metropolitan Hospital, 15562 Athens, Greece; 71st Propedeutic Department of Internal Medicine, Medical School, Aristotle University of Thessaloniki, AHEPA Hospital, 54636 Thessaloniki, Greece; 8Cardiology Clinic, University General Hospital of Heraklion, 70013 Heraklion, Greece; 9Department of Internal Medicine, Medical School, Aristotle University of Thessaloniki, Papageorgiou General Hospital Thessaloniki, 56429 Thessaloniki, Greece; 10Department of Internal Medicine, 424 General Military Training Hospital, 56429 Thessaloniki, Greece; 112nd Propedeutic Internal Medicine Department and Diabetes Research Unit, National and Kapodistrian University of Athens, Attikon University General Hospital, 12462 Athens, Greece; 12Department of Endocrinology, Police Medical Center, 54121 Thessaloniki, Greece; 13Cardiology Clinic, “Tzaneio” General Hospital, 18536 Piraeus, Greece; 14Department of Pediatrics, Medical School, National and Kapodistrian University of Athens, C’ Pediatrics Clinic, Attikon University General Hospital, 12462 Athens, Greece; 151st Propedeutic Department of Medicine, School of Medicine, National and Kapodistrian University of Athens, 17 Agiou Thoma Str., Goudi, 11527 Athens, Greece

**Keywords:** familial hypercholesterolemia, HELLAS-FH, cardiovascular disease, coronary artery disease, diabetes

## Abstract

Familial hypercholesterolemia (FH) and type 2 diabetes mellitus (T2DM) are both associated with a high risk of atherosclerotic cardiovascular disease (ASCVD). Little is known about the prevalence of T2DM and its association with ASCVD risk in FH patients. This was a cross-sectional analysis from the Hellenic Familial Hypercholesterolemia Registry (HELLAS-FH) including adults with FH (n = 1719, mean age 51.3 ± 14.6 years). Of FH patients, 7.2% had a diagnosis of T2DM. The prevalence of ASCVD, coronary artery disease (CAD), and stroke was higher among subjects with T2DM compared with those without (55.3% vs. 23.3%, 48.8% vs. 20.7%, 8.3% vs. 2.7%, respectively, *p* < 0.001). When adjusted for age, systolic blood pressure, smoking, body mass index, hypertension, waist circumference, triglyceride levels, high-density lipoprotein cholesterol levels, and gender, T2DM was significantly associated with prevalent ASCVD [OR 2.0 (95% CI 1.2–3.3), *p* = 0.004]. FH patients with T2DM were more likely to have undergone coronary revascularization than those without (14.2% vs. 4.5% for coronary artery bypass graft, and 23.9% vs. 11.5% for percutaneous coronary intervention, *p* < 0.001). T2DM is associated with an increased risk for prevalent ASCVD in subjects with FH. This may have implications for risk stratification and treatment intensity in these patients.

## 1. Introduction

Familial hypercholesterolemia (FH) is a monogenic, autosomal dominant disorder caused mainly by mutations of the low-density lipoprotein receptor (LDLR), apolipoprotein B (APOB), or proprotein convertase subtilisin/kexin 9 (PCSK9) genes. Impaired LDLR-mediated catabolism of LDL particles leads to lifelong severely high LDL cholesterol (LDL-C) levels and, thus, to a high risk of premature atherosclerotic cardiovascular disease (ASCVD) morbidity and mortality [1]. Indeed, compared with non-FH subjects with LDL-C levels <130 mg/dL, mutation-positive FH individuals with LDL-C levels ≥190 mg/dL have a 22-fold increased ASCVD risk [2]. The prevalence of heterozygous FH (HeFH) in the general population worldwide is about 1 in 300, and that of homozygous FH (HoFH) 1 in 1,000,000, except in regions with high consanguinity rates, such as South Africa, Lebanon, and Quebec, where prevalence is close to 1:100 [3]. Despite its high worldwide prevalence, HeFH remains massively underdiagnosed and undertreated [1]. These patients are usually diagnosed around the fourth decade of life, after 40 years of hypercholesterolemia [4].

Type 2 diabetes mellitus (T2DM) is a highly prevalent disease affecting approximately 463 million individuals worldwide [5]. It is characterized by elevated blood glucose levels or hyperglycaemia due to abnormalities in either insulin secretion or/and insulin action. In addition, altered gut microflora, intestinal dysbiosis and metabolic endotoxemia are considered key mechanisms that seem to be associated with the development of T2DM [6,7]. The abdominal distribution of adipose tissue in T2DM is associated with insulin resistance, hypertension, and lipoprotein abnormalities which constitute the major factors contributing to an increased cardiovascular risk. Atherogenic dyslipidemia in diabetes consists of elevated serum concentrations of triglyceride-rich lipoproteins (TRLs), a high prevalence of small dense LDL, and low concentrations of cholesterol-rich high-density lipoprotein (HDL)-C [8]. The risk of ASCVD is two- to three-fold higher in subjects with T2DM compared with those without, independent of other traditional ASCVD risk factors.

The prevalence of ASCVD in FH subjects with vs. without T2DM has been examined in a limited number of studies [6,9,10]. In contrast to the general population, in which T2DM is a conventional cardiovascular risk factor, the few studies which investigated the association between T2DM and ASCVD risk in FH delivered inconsistent results.

We aimed to explore the prevalence of T2DM and the effect of T2DM coexistence with FH on ASCVD risk in the Hellenic Familial Hypercholesterolemia (HELLAS-FH) Registry.

## 2. Results

A total of 1719 adult patients (873 males) with HeFH were included in the present analysis, 123 of whom (7.2%) had T2DM. Patients with T2DM vs. non-T2DM were significantly older (61.4 ± 11.1 vs. 50.0 ± 14.7 years, *p* < 0.05), had higher body mass index (BMI) [28.7 (25.2–31.8) vs. 26.8 (24.2–29.4) kg/m^2^, *p* < 0.05], and higher prevalence of hypertension (57.7% vs. 24.7%, *p* < 0.05) (Table 1). In addition, patients with T2DM had significantly higher triglyceride levels [151 (120–210) vs. 130 (95–179) mg/dL] and lower HDL-C levels (45 ± 12 vs. 51 ± 15 mg/dL) (Table 2).

Hypolipidemic therapy did not differ significantly between the two groups except for fibrates, which were more commonly prescribed in T2DM patients (6.5% vs. 1.2%, *p* < 0.01) (Table 3). Patients with T2DM were more likely to receive a high-intensity statin (atorvastatin doses of 40 or 80 mg and rosuvastatin doses of 20 or 40 mg are defined as high-intensity statin therapies [11]) compared with non-T2DM subjects (75.3% vs. 63.0%, *p* < 0.05, respectively) (Figure 1). The most common statin prescribed among treated patients in both groups was atorvastatin (51.7% in the T2DM group and 46.1% in the non-T2DM group) at a median dose of 40 mg/day followed by rosuvastatin (30.0% in T2DM group and 39.6% in non-T2DM group) at a median dose of 40 mg/day (Appendix A). The vast majority of FH patients in both groups did not achieve the LDL-C target based on the European Atherosclerosis Society/European Society of Cardiology (EAS/ESC) 2019 guidelines (97.7% of patients with T2DM and 97.1% of non-T2DM patients) (Figure 2).

The prevalence of ASCVD, premature ASCVD, CAD, premature CAD, myocardial infarction (MI), stroke, premature stroke, and peripheral artery disease (PAD) was higher among patients with T2DM compared with non-T2DM (55.3% vs. 23.3%, 48.8% vs. 20.0%, 48.8% vs. 20.7%, 39.0% vs. 15.3%, 43.9% vs. 18.5%, 8.9% vs. 2.7%, 2.4% vs. 0.9%, 7.3% vs. 2.3%, respectively, *p* < 0.05) (Table 4). There was no difference between T2DM and non-T2DM patients regarding a family history of ASCVD (56.9% vs. 53.5%), a family history of CAD (53.7% vs. 47.1%), as well as a family history of premature CAD (35.8% vs. 37.7%).

To assess the effect of T2DM on ASCVD prevalence, a logistic regression analysis was performed. When compared to the null model (*p* < 0.05) the overall model was significant, explaining 4.5% of the variance in ASCVD (Nagelkerke R^2^) and accurately predicting 75.2% of cases. The risk of established ASCVD was four times higher among diabetic patients compared with non-diabetics [unadjusted OR 4.0 (95% CI 2.8–5.9)]. We then carried out the relevant logistic regression analysis taking into account and adjusting for the major ASCVD risk factors age, systolic blood pressure, smoking, BMI, hypertension, waist circumference, triglyceride levels, HDL-C levels, and gender. The mοdel was again significant when compared to the null model (*p* < 0.05), explaining 29.4% of the variance in ASCVD (Nagelkerke R^2^) and accurately predicting 78.6% of cases. The odds of subjects with T2DM having ASCVD were now twice higher than those of non-diabetic subjects [adjusted OR 2.0 (95% CI 1.2–3.3), *p* < 0.05]. In addition, a relevant logistic regression was performed to assess the effect of T2DM, age, systolic blood pressure, BMI, hypertension, waist circumference, triglyceride levels, HDL-C levels, and gender on the likelihood of coronary artery disease (CAD). The overall model was again statistically significant when compared to the null model (*p* < 0.05), describing 31.4% of the variance in ASCVD (Nagelkerke R^2^) and accurately predicting 81.0% of cases. Diabetic patients were almost twice more likely to exhibit CAD than non-diabetics [OR 1.9 (95% CI 1.2–3.2), *p* < 0.05]. Increasing age (*p* < 0.05), hypertension (*p* < 0.05), HDL-C levels (*p* < 0.05), and male gender (*p* < 0.05) were associated with an increased likelihood of exhibiting CAD, but systolic blood pressure, BMI, waist circumference and triglyceride levels were not.

Finally, patients with T2DM were more likely to have undergone a coronary revascularization procedure, such as coronary artery bypass grafting (CABG) or percutaneous coronary intervention (PCI), compared with non-T2DM individuals (14.2% vs. 4.5%, and 23.9% vs. 11.5%, *p* < 0.05, respectively, Appendix A).

## 3. Discussion

In this study, we investigated the prevalence of T2DM in patients with FH in the HELLAS-FH registry. In addition, we showed an adjusted two-fold increase in ASCVD prevalence in FH subjects with T2DM compared with non-T2DM.

There have been controversial findings in the literature regarding the association between dyslipidemia and the risk for T2DM onset. On the one hand, among subjects of middle age from the Framingham Offspring Study, an association between lipid profile parameters and T2DM risk was observed [12]. Specifically, increased levels of triglycerides (>150 mg/dL) and decreased levels of HDL-C (<40 mg/dL in men and <50 mg/dL in women) were positively associated with the onset of T2DM (*p* < 0.05). Similarly, after a 4-year observation period, patients with increased levels of TCHOL (OR 1.18 95%CI 1.05–1.34), LDL-C (OR 1.18 95% CI 1.05–1.33) and triglycerides (OR 1.17 95% CI 1.05–1.29) were positively associated with higher T2DM risk [13]. On the other hand, Mendelian randomization studies have suggested that genetically increased LDL-C levels are associated with decreased risk for T2DM (*p* = 0.0011) [14]. Analogous were the results of a study that examined the association between lipid profile and T2DM risk through mendelian randomization [15]. In this study, a protective role of increased levels of LDL-C (OR 0.79 95% CI 0.71–0.88), HDL-C (OR 0.83 95% CI 0.76–0.90), as well as triglyceride levels (OR 0.83 95% CI 0.72–0.95), was observed regarding T2DM risk.

Another factor that may also play an important role in the association between dyslipidemia patients and T2DM risk is the statin treatment of these patients. Indeed, the management of overt hyperlipidemia inherent in FH patients requires an aggressive hypolipidemic treatment plan for these patients. High-dose as well as high-intensity statins are often co-administered with ezetimibe as well as proprotein convertase subtilisin/kexin type 9 inhibitors (PCSK9i) [16]. The increased risk of new onset T2DM has been established as a previously unrecognized side effect of statins. Indeed, a meta-analysis of randomized statin trials with 91,140 participants showed that statin treatment was associated with a higher risk for new onset T2DM (OR 1.09; 95% CI 1.02–1.17) [17]. Therefore, an increased risk of T2DM among aggressively statin-treated FH patients may have been expected. However, data from the SAFEHEART (Spanish Familial Hypercholesterolaemia Cohort Study) study showed that extended high-dose statin treatment was not associated with a higher risk of T2DM in FH patients [18].

It has been suggested that the prevalence of T2DM is lower in FH compared with non-FH subjects: 1.75% in FH patients from the national Dutch screening program vs. 2.93% in unaffected relatives [*p* < 0.001; odds ratio (OR), 0.62 (95% CI, 0.55–0.69)] [19], and 5.94% in patients with FH from the Spanish Dyslipidemia Registry compared with 9.44% in the general population (*p* < 0.001; OR: 0.61, 95% CI 0.49–0.76) [20]. The prevalence of T2DM in the general population in Greece was 11.6% in the recent National Survey of Morbidity and Risk Factors (EMENO), as compared with 7.2% in the HELLAS-FH. In EMENO the median age of participants was comparable to the HELLAS-FH registry (47.9 vs. 51.0 years, respectively) [21]. Two other studies in Greek adults found that the age-adjusted T2DM prevalence was 11.8% and 10.4%, respectively [22,23]. Previous studies in Greek groups have also reported diabetes prevalence estimates ranging from 2.4% to 9.5% [24]. Twenty years ago (2001–2002), the ATTICA study, a nationwide study, assessed the diabetes prevalence in men (7.9%), as well as in women (6.0%) [25]. The prevalence of T2DM in HELLAS-FH is higher when compared with those reported by the Canadian FH registry (5.0%) [26], and the Spanish SAFEHEART registry (5.9%) [20], but lower compared to the American Cascade Screening for Awareness and Detection (CASCADE)-FH registry (13.0%) [27]. Of note, the prevalence of T2DM in the HELLAS-FH registry is much higher in comparison to the Dutch-FH registry (1.75%) [19].

This decreased prevalence of T2DM in HELLAS-FH could be related to the possible healthier lifestyle of FH subjects, lower BMI, and close contact with health care providers [28]. Additionally, it has been suggested that the LDLR plays a role in the toxicity of cholesterol on the beta cells, i.e., impaired LDLR function in FH could protect beta cells against the possible deleterious LDL particle entry [28]. LDLR mutations in FH patients may lower the risk of T2DM as well as protect from the diabetogenic effect of statins [29]. It has been found that the more severe the LDL-C metabolic defect, the less the T2DM prevalence is. Among the participants in the national Dutch screening program who underwent DNA testing for FH, those who were LDL receptor-negative mutation carriers had less T2DM (1.12%) than those who were LDL receptor-deficient (1.44%) or were carriers of apoB mutations (1.91%) [19]. However, a recent study in FH patients in Gran Canaria demonstrated a high prevalence of T2DM especially among carriers of the mutation *p*. [Tyr400_Phe402del] [30]. Indeed, these patients had more often T2DM compared with patients having other mutations (25% vs. 4%, *p* = 0.045%). Similarly, two other studies also showed a high prevalence of T2DM among 302 Serbian FH patients (22.8%) [31] and 289 Chinese FH patients (20.1%) [32]. Nevertheless, both studies included patients without genetic confirmation of FH, based only on Dutch Lipid Clinic Criteria [33].

Differences among FH registries may explain the different prevalence of T2DM in FH subjects. While many registries are based on molecular FH diagnosis, as we are beginning to see with the CASCADE-FH registry, in several others, including the HELLAS-FH, clinical criteria were mainly applied. It has been reported that in studies showing a high prevalence of T2DM, more than 20%, mainly included subjects with a clinical diagnosis of FH [31,32]. For this reason, more registries should incorporate molecular diagnosis for improved identification of FH patients. Additionally, in this direction, the EAS Familial Hypercholesterolaemia Studies Collaboration (FHSC) is an initiative of international stakeholders to establish a global FH registry, expand knowledge regarding FH and upgrade the care provided to FH patients [34]. Moreover, the diagnosis of T2DM is in many FH registries, such as the HELLAS-FH registry, self-reported [19], while others record abnormal biochemical data or antidiabetic drug prescriptions [20]. The FHSC recently reported that the overall prevalence of T2DM is 5.0%, and 1.1% among subjects younger than 40 years [28]. This is, in part, to be expected, since T2DM prevalence increases with rising age [35]. In this context, the age of subjects in the Dutch-FH registry was much lower compared with the HELLAS-FH registry (37.6 vs. 51.0 years), which may partly explain the much lower prevalence of T2DM in the Dutch-FH registry [19].

ASCVD is the leading cause of morbidity and mortality in individuals with T2DM [5]. As it has been recently demonstrated, compared with T2DM controls from the general Swedish population, subjects with both T2DM and FH have increased cardiovascular mortality [hazard ratio (HR) 2.40, 95% confidence interval (CI) 2.19–2.63] and risk of ASCVD events (HR 2.73, 95% CI 2.58–2.89) [36]. Patients with T2DM in HELLAS-FH had a higher prevalence of ASCVD compared with non-T2DM (55.3% vs. 23.3%, *p* < 0.05). Part of this difference may be explained by differences in age (61 vs. 51 years) and increased prevalence of additional ASCVD risk factors, which were more common in T2DM vs. non-T2DM subjects (Table 1). Following adjustment for age, systolic blood pressure, smoking, BMI, hypertension, waist circumference, triglyceride levels, HDL-C levels, and gender, the association between T2DM and prevalent ASCVD attenuated but remained significant (adjusted OR 2.0 (95% CI 1.2–3.3), *p* < 0.05). Additionally, FH patients with T2DM had a history of coronary revascularization to a greater percentage than those without, a finding that might indicate more severe ASCVD. As a result, it can be inferred that the deleterious effects of T2DM on ASCVD risk remain relevant even in the setting of patients with an already high ASCVD risk. Indeed, we showed that T2DM doubles the risk of established ASCVD among clinically diagnosed FH subjects.

Evidence regarding the association between T2DM and ASCVD in patients with FH is scarce. Multiple logistic regression analysis in a Dutch-FH cohort (n = 14,283 subjects) identified T2DM as an independent risk factor for ASCVD (unadjusted OR: 1.37, 95% CI, 1.03–1.82, *p* = 0.03) [37]. Data from the Spanish Arteriosclerosis Society FH Registry (n = 1724) also confirmed a positive association between T2DM and ASCVD (adjusted OR: 2.01, 95% CI, 1.18–3.43; *p* = 0.01) [20]. In a cross-sectional analysis of 1295 HeFH adults from the CASCADE-FH registry from 11 US lipid clinics, T2DM was one of the factors that were associated with prevalent CAD (adjusted OR: 1.74, 95% CI 1.08–2.82) [27]. Similarly, a retrospective, multi-center, cohort study, conducted in the Netherlands and including 2400 FH patients, proved that T2DM along with male gender, hypertension, smoking, HDL-C, and lipoprotein (a) [Lp (a)] levels are independent ASCVD risk factors [38]. A study in 1050 Japanese FH patients showed that T2DM is independently associated with incident major adverse cardiovascular events (HR 1.81, 95% CI 1.12–2.25; *p* = 0.047) [39]. Another recent Chinese study in a HeFH cohort concluded that T2DM is an independent predictor of CAD severity. Patients with T2DM were at a significantly greater risk of hard cardiovascular endpoints [multivariate adjusted HR 2.32, 95% CI 1.02–4.84; *p* =0.025]. Of note, patients with T2DM and good glucose control (HbA1c < 7.0%) were at a lower risk of hard endpoints compared with those with poor glucose control (HbA1c ≥ 7.0%, HR 0.08, 95% CI 0.01–0.56; *p* = 0.011). [40]. Moreover, a meta-analysis of 27 studies, including over 41 thousand individuals and 6629 cardiovascular events, concluded that T2DM, along with smoking and hypertension represent more than a quarter of ASCVD risk in FH patients [41].

These findings could not be replicated in certain FH cohorts, such as those from Canada [9], USA [42], Brazil [43], Australia [44], Greece [45], and Spain [46]. Specifically, T2DM was no longer significantly associated with the presence of ASCVD after adjustment for other covariates in those studies. As may be assumed, the contribution of T2DM to the cardiovascular risk may be less important in FH patients than in the general population. Nevertheless, all these cohorts either evaluated a lower number of FH patients with T2DM, or the prevalence of T2DM was very low, possibly limiting the power to detect an association between T2DM and ASCVD and underestimating the role of T2DM as a cardiovascular risk factor in the FH population. For these reasons, when assessing the individual risk of an FH patient with T2DM, it is safer to take into consideration other parameters related to the condition as well, such as the duration of T2DM and target organ damage [47].

Finally, an extremely high proportion of patients were not on LDL-C target (>97% for both groups). LDL-C targets are often not achieved in FH patients. Most patients will eventually need a third or even a fourth lipid-lowering medication to achieve contemporary LDL-C targets [48]. As we have previously shown, a very low percentage of patients in the HELLAS-FH registry reach new LDL-C targets (2.7%), even if on maximum intensity statin/ezetimibe treatment (5.9%) [48]. Similar observations have been reported from the PLANET registry for patients in the Czech Republic and Slovakia (7.2% on LDL-C target of <70 mg/dL) [49], and from the SAFEHEART registry (1.1% of ASCVD patients on LDL-C target, 11% of patients achieved LDL-C targets receiving maximum hypolipidemic treatment) [50]. This inadequate attainment of LDL-C treatment goals may be due to the extremely high baseline LDL-C levels, which are difficult to decrease enough with currently available hypolipidemic agents. Additionally, many FH patients are not treated with the combination of a high-intensity statin with ezetimibe, and besides, many heterozygotes with certain mutations will not be able to achieve the LDL-C goal with maximally tolerated, high-dose statin therapy even when combined with ezetimibe [27,50]. Of note, a large number of subjects were registered in the HELLAS-FH before the publication of the last ESC/EAS guidelines [11] which recommend more stringent LDL-C goals than those from 2016 [51].

## 4. Study Strengths and Limitations

In contrast to other FH cohort studies, we used only ASCVD endpoints that were explicit and well defined, such as MI, CABG, PCI, stroke, CAD, and PAD. This let us avoid any misconstructions regarding ambiguous diagnoses, such as angina, transient ischemic attack, and positive exercise stress tests. Although this excluded some individuals from the ASCVD group, it makes our results more robust.

There are several limitations in this study. First, its cross-sectional design does not allow us to establish a causal role of T2DM with ASCVD risk. Another limitation is the lack of information on T2DM duration, the status of glucose control, and the type of antidiabetic medications used [52]. Additionally, data were retrieved from patient medical files and laboratory studies performed in local laboratories. Furthermore, patients with T2DM were significantly older and had higher BMI, blood pressure, and triglyceride levels. Even after multiple statistical adjustments, residual confounding may remain. It would be better if apolipoprotein B (ApoB) and E (ApoE) were included in the analysis. ApoB may be useful in predicting CVD risk in states of insulin resistance [53]. Additionally, ApoB is highly associated with T2DM and may be a risk factor for T2DM [54]. In addition, ApoE is described as a major ligand for LDL receptors with a role in cholesterol metabolism and cardiovascular disease [55]. However, we do not have ApoB and ApoE levels available for most of the patients. What is more, the use of clinical criteria for the diagnosis of FH may have allowed some patients with polygenic hypercholesterolemia to be misclassified as FH. Undoubtedly, molecular diagnosis adds confidence to the diagnosis and helps to avoid misclassifications.

## 5. Materials and Methods

### 5.1. Study Design

The HELLAS-FH registry design and rationale have been previously described [56,57,58]. In brief, HELLAS-FH utilizes a cluster of sites distributed in many Greek cities. Patients with FH, after signing an informed consent form, are enrolled in an online database. The diagnosis of FH is clinical, based on the Dutch Lipid Clinic Network (DLCN) criteria which have been suggested to have an 85% agreement rate with the genetic diagnosis [33]. Patients with a DLCN score ≥3 (at least possible FH) are eligible for enrolment into the registry. This is in accordance with the latest ESC/EAS consensus statement on the clinical diagnosis of FH [11]. Patients with homozygous FH were excluded from the present analysis. LDL-C levels were calculated by the Friedewald formula [LDL-C = total cholesterol (TCHOL) − triglycerides/5 − HDL-C] [59].

All laboratory measurements were performed in each site’s hospital laboratory using standard methods after an overnight fast. ASCVD includes CAD (with or without coronary revascularization procedures such as CABG and/or PCI), stroke, and PAD. ASCVD events were patient reported and additionally validated by official copies of relevant medical history notes. The European Society of Hypertension (ESH) guidelines [60] and the American Diabetes Association Standards of Medical Care in Diabetes [61] were used for the diagnosis of hypertension and diabetes, respectively. Smoking status was considered positive for both past and current smokers.

### 5.2. Statistical Analysis

The Kolmogorov–Smirnov test was used to test continuous variables for lack of normality. Values are expressed as mean ± standard deviation (SD) and median (interquartile range—IQR) for parametric and non-parametric variables, respectively. Characteristics of the study participants are presented as frequencies and percentages for categorical variables and the Chi-squared test was used for their comparisons. Continuous variables were compared using the Student’s t-tests or the Mann–Whitney U test depending on the variable distribution. The effect of various parameters on a certain outcome was assessed using binary logistic regression analysis adjusting also for the major ASCVD risk factors (age, systolic blood pressure, smoking, BMI, hypertension, waist circumference, triglyceride levels, HDL-C levels, and gender). A *p* < 0.05 was considered significant. The Statistical Package for the Social Sciences (SPSS) 21.0 (SPSS Inc, Chicago, IL, USA) was used to perform study analyses.

## 6. Conclusions

T2DM in patients with FH is not as rare as was previously thought. Furthermore, the coexistence of FH and T2DM is associated with a two-fold increased risk of ASCVD. As for the LDL-C goal attainment, defined as a treated LDL-C < 70 mg/dL or a ≥50% LDL-C reduction, it remains low. Efforts to mitigate risk must begin much earlier in life to be effective. Early diagnosis of FH and administration of hypolipidemic therapy, early up-titration to high-intensity statins in combination with other lipid-lowering agents, specific targeting and comprehensive management of modifiable traditional cardiovascular risk factors, as well as periodical screening for T2DM in parallel with lifestyle modification, are required for the care of this high-risk group of patients. In addition, further studies are still needed to elucidate whether T2DM is associated with incident ASCVD events in FH and to establish the best therapeutic approach to diminish the high CVD risk in the diabetic FH population.

## Figures and Tables

**Figure 1 pharmaceuticals-16-00044-f001:**
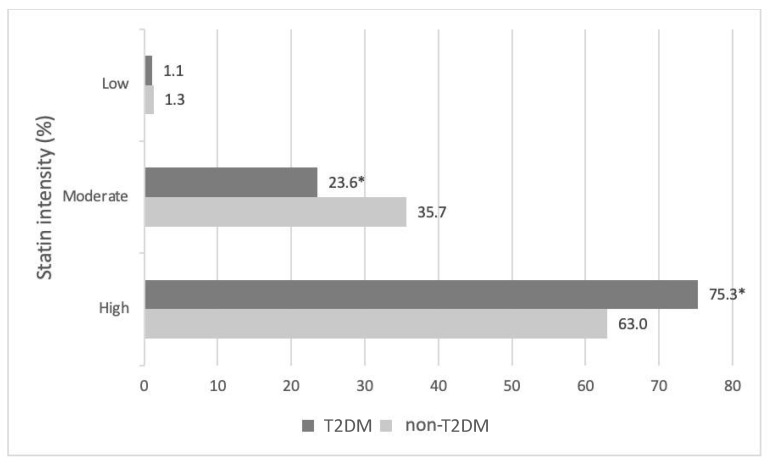
Statin intensity stratified by T2DM status. * *p* < 0.05 vs. non-T2DM.

**Figure 2 pharmaceuticals-16-00044-f002:**
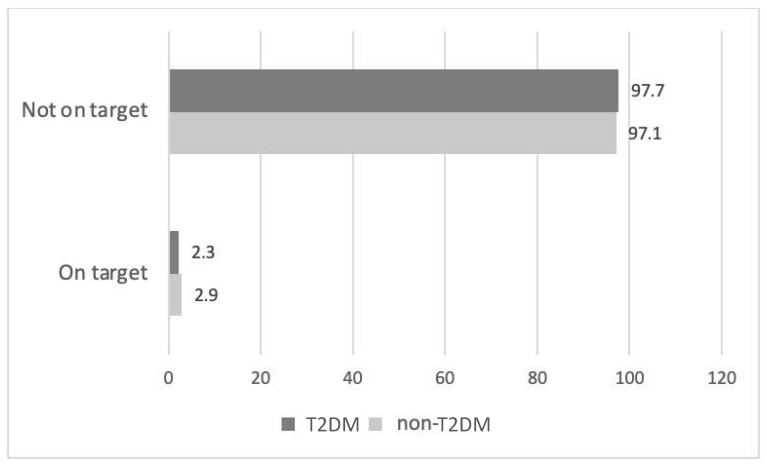
Target achievement among treated patients stratified by T2DM status based on EAS/ESC 2019 guidelines. ESC: European Society of Cardiology; EAS: European Atherosclerosis Society.

**Table 1 pharmaceuticals-16-00044-t001:** Baseline characteristics of FH patients in the HELLAS-FH registry.

			T2DM	Non-T2DM	*p* Value (vs. Non-T2DM)
**Number of patients**			123	1596	
**Gender (male/female)**			70/53	803/793	
**Age at registration (years)**			61.4 ± 11.1	50.0 ± 14.7	<0.05
**Age at diagnosis (years)**			53.9 ± 3.0	43.4 ± 15.9	<0.05
**DLCN score**			5 (4–7)	5 (4–8)	NS
**Systolic blood pressure (mmHg)**			132 ± 14	128 ± 14	<0.05
**Diastolic blood pressure (mmHg)**			78 ± 10	77 ± 9	NS
**Heart rate (bpm)**			73 ± 10	74 ± 10	NS
**Prevalence of distinctive clinical findings (%)**		** *Corneal arcus below* ** ** *the age of 45 years* **	7.3%	7.2%	NS
		** *Tendon xanthomas* **	7.3%	5.4%	NS
		** *Xanthelasma* **	5.7%	5.8%	NS
**Body mass index (kg/m^2^)**			28.7 (25.2–31.8)	26.8 (24.2–29.4)	<0.05
**Hypertension (%)**			57.7	24.7	<0.05
**Waist circumference**	** *Male (cm)* **		100 (88–110)	95 (88–103)	NS
	** *Female (cm)* **		97 (87–104)	89 (80–98)	<0.05
	** *Men >102 cm (%)* **		32.9	24.4	NS
	** *Women >88 cm (%)* **		58.5	46.2	NS
**Smokers (%)**	** *Active* **		22.0	25.6	NS
	** *Former* **		17.1	9.3	<0.05
	** *Never* **		61.0	65.2	NS

FH: Familial hypercholesterolemia; T2DM: Type 2 diabetes; DLCN: Dutch Lipid Clinic Network; NS: non-significant.

**Table 2 pharmaceuticals-16-00044-t002:** Lipid profile of FH patients in the HELLAS-FH registry.

	T2DM		Non-T2DM		*p* Value (vs. Non-T2DM)
Parameter	At Diagnosis (n = 123)	On Treatment (n = 92)	At Diagnosis (n = 1596)	On Treatment (n = 1106)	
Total cholesterol (mg/dL)	319 ± 95	205 ± 57	328 ± 88	209 ± 54	NS
Triglycerides (mg/dL)	151 (120–210)	130 (101–197)	130 (95–179)	108 (79–150)	<0.05
HDL-C (mg/dL)	45 ± 12	47 ± 18	51 ± 15	52 ± 16	<0.05
Non-HDL-C (mg/dL)	272 ± 96	157 ± 54	277 ± 89	156 ± 54	NS
LDL-C (mg/dL)	236 ± 94	126 ± 49	247 ± 87	132 ± 50	NS
Lp (a) (mg/dL)	28 (13–74) ^†^	28 (13–72) ^†^	27 (18–91) ^††^	26 (10–61) ^††^	NS

FH: Familial hypercholesterolemia; T2DM: Type 2 diabetes; HDL-C: High-density lipoprotein cholesterol; LDL-C: low-density lipoprotein cholesterol; Lp (a): Lipoprotein (a); NS: non-significant. Data are presented as mean ± standard deviation or median (interquartile range) for parametric and non-parametric variables, respectively; ^†^: data available for 20 patients; ^††^: data available for 269 patients.

**Table 3 pharmaceuticals-16-00044-t003:** Hypolipidemic treatment among treated FH patients in the HELLAS-FH registry stratified by T2DM status.

Treatment (%)	T2DM	Non-T2DM	*p* Value (vs. Non-T2DM)
Statin	94.6	97.7	NS
Ezetimibe	42.4	48.4	NS
Statin + ezetimibe	39.1	47.4	NS
PCSK9i	7.6	5.3	NS
Fibrate	6.5	1.2	<0.05
Bile acid sequestrants	0.0	1.6	NS
n3 fatty acids	2.2	1.5	NS
Sterols/stanols	0.0	0.4	NS

FH: Familial hypercholesterolemia; T2DM: Type 2 diabetes; PCSK9i: Proprotein convertase subtilisin/kexin type 9 inhibitors; NS: non-significant.

**Table 4 pharmaceuticals-16-00044-t004:** ASCVD prevalence in the HELLAS-FH registry stratified by T2DM status.

Prevalence of ASCVD	T2DM Patients	Non-T2DM Patients	*p* Value (vs. Non-T2DM)
Total ASCVD	55.3%	23.3%	<0.05
Premature total ASCVD	48.8%	20.0%	<0.05
CAD	48.8%	20.7%	<0.05
MI	39.0%	15.3%	<0.05
Premature CAD	43.9%	18.5%	<0.05
Stroke	8.9%	2.7%	<0.05
Premature stroke	2.4%	0.9%	<0.05
PAD	7.3%	2.3%	<0.05
Premature PAD	2.4%	0.9%	NS

T2DM: Type 2 diabetes mellitus; ASCVD: atherosclerotic cardiovascular disease; CAD: coronary artery disease; MI: myocardial infarction; PAD: peripheral arterial disease; NS: non-significant.

## Data Availability

Data is contained within the article and Appendix A.

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
