# Peer review of "Prevalence of Diabetes and Its Association with Atherosclerotic Cardiovascular Disease Risk in Patients with Familial Hypercholesterolemia: An Analysis from the Hellenic Familial Hypercholesterolemia Registry (HELLAS-FH)"

_pharmaceuticals, 2022, doi:10.3390/ph16010044_

Round 1

Reviewer 1 Report

1)     From line 68-73, the authors have discussed the background of T2DM but in certain only. This current information for T2DM provided is insufficient and authors are suggested to add more data about the pathogenesis of T2DM. 

2)     In introduction section, the authors have missed the information and association of atherosclerosis with their hypothesis. It should be included in this part. For this, the following references may please be considered: J Food Biochem. 2020; 44(11): e13449. https://doi.org/10.1111/jfbc.13449; Crit Rev Immunol. 2019; 39(4): 223-237. https://doi.org/10.1615/CritRevImmunol.2019030614

3)     In the manuscript, the authors have used T2D and non-T2D. This term should be written as T2DM not T2D.

4)     In table 4, the authors have mentioned the prevalence of ASCVD in the HELLAS-FH. In current format, this table is quite confusing. It should be revised and its correlation and association with other factors should be considered.

5)     In manuscript, the authors have used different values to represent the statistically significant difference. I suggest, only one value should be used instead of different ones.

6)     Discussion needs major revision as in its current format, it is very difficult to understand what do the authors want to discuss.

7)     Have the authors considered ASCVD in their study?

8)     Similarly, the conclusion of this study in its present form does not reflect the key findings of the study. It should reflect the overall key findings in accordance with future perspectives and its impact on the society.

There are several grammatical mistakes and syntax errors. The whole manuscript needs critical revision to remove all the grammatical mistakes and syntax errors.

Author Response

  1. From line 68-73, the authors have discussed the background of T2DM but in certain only. This current information for T2DM provided is insufficient and authors are suggested to add more data about the pathogenesis of T2DM. 

We thank the reviewer for the comment. This was added to text: “It is characterized by elevated blood glucose levels or hyperglycemia due to abnormalities in either insulin secretion or/and insulin action. In addition, altered gut microflora, intestinal dysbiosis and metabolic endotoxemia are considered key mechanisms that seem to be associated with the development of T2D [6,7]. The abdominal distribution of adipose tissue in T2D is associated with insulin resistance, hypertension, and lipoprotein abnormalities which constitute the major factors contributing to an increased cardiovascular risk. Atherogenic dyslipidemia in diabetes consists of elevated serum concentrations of triglyceride-rich lipoproteins (TRLs), a high prevalence of small dense LDL, and low concentrations of cholesterol-rich high-density lipoprotein (HDL)2-C [8].” (page 5; lines 91-97 and page 6; lines 98-100).

  1. In introduction section, the authors have missed the information and association of atherosclerosis with their hypothesis. It should be included in this part. For this, the following references may please be considered: J Food Biochem. 2020; 44(11): e13449. https://doi.org/10.1111/jfbc.13449; Crit Rev Immunol. 2019; 39(4): 223-237. https://doi.org/10.1615/CritRevImmunol.2019030614

The Reviewer raises a very important point. We included this sentence along with the appropriate references: “In addition, altered gut microflora, intestinal dysbiosis and metabolic endotoxemia are considered key mechanisms that seem to be associated with the development of T2D [6,7].” (page 5; lines 93-95 and new references 6 and 7).

  1. In the manuscript, the authors have used T2D and non-T2D. This term should be written as T2DM not T2D.

According to the reviewer’s consideration, we corrected the term throughout the manuscript.

  1. In table 4, the authors have mentioned the prevalence of ASCVD in the HELLAS-FH. In current format, this table is quite confusing. It should be revised and its correlation and association with other factors should be considered.

We have now simplified the format of table 4 in order to improve its readability (page 11; lines 178-182). Moreover, the correlation and associations of additional ASCVD risk factors with the presence of ASCVD in patients with and without T2DM are separately explored using a multivariate logistic regression analysis model (page 12; lines 183-204).

  1. In manuscript, the authors have used different values to represent the statistically significant difference. I suggest, only one value should be used instead of different ones.

We now only use one value (p<0.05) in order to identify statistically significance.

  1. Discussion needs major revision as in its current format, it is very difficult to understand what do the authors want to discuss.

We thank the reviewer for the comment. We have now significantly restructured the discussion section so that the reader can more clearly identify the major discussion topics. In brief, we first discuss the association of hyperlipidemia with T2DM as well as the associations of statin treatment with T2DM onset. We then comment on the T2DM prevalence differences between patients of the general population as compared with FH subjects. Subsequently we propose possible explanations for differences in T2DM prevalence between FH and non-FH patients as well as insights regarding differences of T2DM prevalence among various FH registries. Moreover, we discuss the association between ASCVD and T2DM in FH populations as well as compare our findings with other registries. Finally, we comment on the effect of T2DM regarding the achievement of LDL-C targets in FH patients.

  1. Have the authors considered ASCVD in their study?

The aim of the study was to evaluate, apart from the prevalence of diabetes, its association with ASCVD in patients with familial hypercholesterolemia (FH) and we finally found that the coexistence of FH and T2DM is associated with a 2-fold increased risk of prevalent ASCVD. (page 12; lines 183-204).

  1. Similarly, the conclusion of this study in its present form does not reflect the key findings of the study. It should reflect the overall key findings in accordance with future perspectives and its impact on the society.

We thank the reviewer for the comment. We have already reported the conclusions regarding the prevalence of diabetes and the risk for prevalent ASCVD in patients with FH, which were the aims of this study. We’ve also highlighted the low rates of LDL-C goal attainment. In order to emphasize the importance of prevention and effective management of T2DM in FH patients we added a few more sentences to the text (“Efforts to mitigate risk must begin much earlier in life to be effective.”,  “periodical screening for T2DM”). (page 22; lines 435-436 and page 23; line 439).

Reviewer 2 Report

The manuscript submitted by Boutari et al aimed to evaluate the prevalence of diabetes and its association with ASCVD among patient with heterozygous familial hypercholesterolemia included in the Hellenic Familial Hypercholesterolemia Registry. It is an interesting topic and adds data on the above-mentioned topics in the Greek population given the limited data on this subject. It also provides information on the level of achievement LDL targets in this population.

The manuscript is well written and easy to follow. The objective is clearly defined, the methodology seems appropriate although is not fully described. The Results are well structured, and the conclusions are supported by the results.

I have a few comments that are listed below:

It would add to the quality of the manuscript to have all p-values for comparison between groups listed in all tables.

Please state how was defined the intensity of statin therapy and add the numbers for the % of patients that were treated with a combination of hypolipidemic drugs.

For the logistic regression, please describe what was included in the model. As the traditional risk factors were used for correction this should be clarified.

Please state in the Methods section the tests used to compare continuous variables.

Please define all abbreviations at their first use in the main text. For example, ASCVD.

Author Response

According to Reviewer 2 comments:

  1. It would add to the quality of the manuscript to have all p-values for comparison between groups listed in all tables.

We added the p-values to the tables of the manuscript.

  1. Please state how was defined the intensity of statin therapy and add the numbers for the % of patients that were treated with a combination of hypolipidemic drugs.

Thank you for this recommendation. We added the definition: (atorvastatin doses of 40 or 80 mg and rosuvastatin doses of 20 or 40 mg are defined as high intensity statin therapies [11]). (page 8; lines 137-138 and reference 11). We report the percentage of T2DM and non-T2DM patients (39.1% and 47.4, respectively). (page 9; lines 148-152; Table 3)

  1. For the logistic regression, please describe what was included in the model. As the traditional risk factors were used for correction this should be clarified.

We have described that in the results and we added that to the methods section also. (page 12; lines 189-190 and page 22; lines 427-429).

  1. Please state in the Methods section the tests used to compare continuous variables.

We now explain in the Methods section that continuous variables were compared using the Student's t-tests or the Mann–Whitney U test depending on variable distribution. (page 22; lines 424-425).

  1. Please define all abbreviations at their first use in the main text. For example, ASCVD.

We thank the reviewer for the suggestion. We adjusted that.

Reviewer 3 Report

This great paper shows an important link between FH, T2DM and CV disease. I have some remarks:

- It is a good sign that non-HDL-C was included in the analysis, given the shortcomings of LDL-C in guidelines, but ApoB and ApoE could be even more important according to recent research, and the authors should discuss this.

- Certain FH mutations render some medications (such as statins) useless, causing an inherently bigger CV risk, and this is not mentioned in the text.

- T2DM is essentially an exclusion diagnosis, and can also be divided depending on either major insulin resistance or major defect in insulin secretion. Is anything of this information present in the registry?

Author Response

According to Reviewer 3 comments:

  1. It is a good sign that non-HDL-C was included in the analysis, given the shortcomings of LDL-C in guidelines, but ApoB and ApoE could be even more important according to recent research, and the authors should discuss this.

We agree with the comment of the reviewer, however we do not have ApoB and ApoE levels available for the majority of the patients.

We added this to the text: “It would be better if apolipoprotein B (ApoB) and E (ApoE) were included in the analysis. ApoB may be useful in predicting CVD risk in states of insulin resistance [55]. Also, ApoB is highly associated with T2DM and may be a risk factor for T2DM [56]. In addition, ApoE is described as a major ligand for LDL receptors with a role in cholesterol metabolism and cardiovascular disease [57]. However, we do not have ApoB and ApoE levels available for most of the patients.”. (page 20; lines 387-392).

  1. Certain FH mutations render some medications (such as statins) useless, causing an inherently bigger CV risk, and this is not mentioned in the text.

We thank the reviewer for this recommendation. We added that to the text: “and besides, many heterozygotes with certain mutations will not be able to achieve LDL-C goal with maximally tolerated, high-dose statin therapy even when combined with ezetimibe [33,52].” (page 19; lines 367-368 and page 20; line369).

  1. T2DM is essentially an exclusion diagnosis, and can also be divided depending on either major insulin resistance or major defect in insulin secretion. Is anything of this information present in the registry?

Unfortunately, the diagnosis of T2DM in the HELLAS-FH registry is self-reported and we do not have any data concerning insulin resistance or insulin secretion.

Reviewer 4 Report

Familial hypercholesterolemia (FH) is the most common genetic disorder of lipoprotein metabolism. Recent research has shown that diabetic FH patients represent a high-risk group for CVD risk, most likely since diabetic subjects have many concomitant cardiometabolic risk factors. In this regard, the present study explores an area of new research.

To improve the manuscript, I recommend:

-          Lines 70-88 should be in the Discussion section. In the Introduction section, the authors should highlight the importance of the topic, and provide an up-to-date and well-focused literature review stating the main and specific objectives of the contribution and the perspectives of the study results.

-  Please define the aim of the study in the Introduction section.

- In lines 73, 164, and 287, please add references to the data presented.

Author Response

According to Reviewer 4 comments:

  1. Lines 70-88 should be in the Discussion section. In the Introduction section, the authors should highlight the importance of the topic, and provide an up-to-date and well-focused literature review stating the main and specific objectives of the contribution and the perspectives of the study results.

We agree with the reviewer’s comment and we transferred this part to the Discussion. (page 17; lines 301-305, page 13; lines 215-227 and page 14; lines 228-230).

  1. Please define the aim of the study in the Introduction section.

We transferred the aim to a separate paragraph in the end of the Introduction section. (page 6; lines 107-109).

  1. In lines 73, 164, and 287, please add references to the data presented.

We would like to thank the reviewer for the suggestion. We added the appropriate references. (page 17; line 305; reference 39, page 14; line 236; reference 18, page 21; line 408; reference 63).

Round 2

Reviewer 1 Report

Authors have revised their manuscript significantly and addressed all the comments in an appropriate way. I have no further comments.